# Study on Spline Stress of Separator Plates in a Wet Multi-Plate Clutch

**DOI:** 10.3390/ma17123039

**Published:** 2024-06-20

**Authors:** Biao Ma, Xiaobo Chen, Changsong Zheng, Liang Yu, Qin Zhao, Weichen Lu

**Affiliations:** School of Mechanical Engineering, Beijing Institute of Technology, Beijing 100081, China; mabiao@bit.edu.cn (B.M.); 3120205194@bit.edu.cn (X.C.); yuliang@bit.edu.cn (L.Y.); yudewuhou@outlook.com (Q.Z.); 3120210312@bit.edu.cn (W.L.)

**Keywords:** clutch, separator plate, spline tooth, stress distribution, fracture

## Abstract

The spline teeth fracture of separator plates in wet multi-plate clutches compromises driving safety and the vehicle’s lifespan. Tooth fracture is mainly caused by stress concentration at the tooth root and uneven circumferential load distribution. This paper considers parameters such as torque, teeth count, tooth profile, and misalignment errors, establishing the corresponding finite element (FE) model to analyze the impact of the above-mentioned parameters on the strength of the separator plates. Analysis under even and biased load circumstances demonstrated that an optimum tooth count and profile can significantly increase the strength of the separator plates, offering advice for the optimized design of wet multi-plate clutch separator plates.

## 1. Introduction

The primary function of a clutch is to transmit and interrupt the torque flow from the engine to the transmission system, which is crucial for enabling smooth vehicle start-up, facilitating gear shifts, and managing power output during operation [1,2]. As is shown in Figure 1, the wet multi-plate clutch is mainly composed of two friction elements, the separator plate and the friction plate, both of which exhibit the characteristics of an annular thin plate structure and are stacked in an alternating manner. The friction plates’ internal splines are connected to the transmission gear shaft’s external splines (driving part), whereas the separator plates’ external splines are attached to the cylinder liner’s keyway (driven part). The clutch uses the viscous shearing of the lubricating fluid and the relative sliding contact between the friction elements to transmit torque.

When the clutch is operating, the separator plate experiences various forces including axial pressure, frictional forces (circumferential and axial), viscous shear forces, interaction forces from the cylinder liner, and thermal stress (resulting from uneven heating). The primary failure forms encompass conical buckling, wear, tooth fracture or crack, and tooth flank failure [3]. According to statistics, around 20–30% of the failures could be attributed to tooth fracture or crack in marine wet clutch [4]. Figure 1 presents a schematic diagram of a typical wet multi-plate clutch and its separator plates’ tooth fracture and crack.

The premature failures, such as the cracks and fractures of the separator plate’s teeth, can greatly diminish the durability of the clutch, hence substantially reducing the service life of the integrated transmission system and even the entire vehicle [5], which have long been a bottleneck restricting the development of vehicle transmission systems towards transmitting greater mechanical torque and achieving higher power density [6]. Consequently, a comprehensive examination of the spline stress is required. To date, many scholars have conducted research on the fracture mechanism of spline teeth.

The tooth fracture is closely related to its structure. Sinha et al. [7] found that tooth fractures usually result from root fillet fatigue cracks. Leen et al. [8,9] and Ding et al. [10] added that stress concentration at the tooth base caused fatigue cracks. Li [11] found that tooth profile deviations and pitch errors could create severe contact and bending loads. Tjernberg [12] observed that pitch errors increased tooth root stress by approximately 26–36% relative to ideal pitching. Chen et al. [13] found that the axial engagement length affected the bending and contact strength and the stress concentration coefficients of axial load distribution are between 3.5 and 5.1 [14,15,16,17,18]. Finally, it was found that increasing spline shaft diameter, teeth count, and tooth thickness could strengthen spline couplings [19,20,21].

The spline coupling between the separator plate and the cylinder liner, as a multi-tooth meshing pair, exhibits uneven contact characteristics caused mainly by manufacturing faults, assembly flaws, and wear, which may result in overload fracture.

Manufacturing faults are unavoidable. Chase et al. [22] discovered that even when made with precision gear hobbing processes, only a fraction of the teeth could actually engage. Kahraman [23] discovered that tooth-side clearance and positioning errors caused uneven load distribution in the circumferential direction. Hong et al. [24,25] observed that for a given manufacturing mistake, the load distribution coefficient of the crucial teeth fell within a known particular range, and the splines with tight manufacturing tolerances had better inter-tooth load distribution properties. Wink et al. [26] observed that when the tooth-side clearance increases due to machining defects, so does the maximum inter-tooth load distribution coefficient.

Assembly faults are also unavoidable. Misalignment is the most prevalent assembly fault, which is of three types: angular, radial, and combined misalignment. Figure 2 shows the illustrative diagrams of the three kinds of misalignment.

Researchers discovered that misalignment reduced the count of meshing teeth [27]. When the misalignment was minimal, the load distribution among the teeth followed a trigonometric distribution pattern, and under combined misalignment situations, radial misalignment had the greatest impact on circumferential load distribution [28], whereas angular misalignment had the greatest influence on maximum contact stress on spline teeth [29]. In misaligned situations, the splines contacted just two-thirds of the tooth pairs [30]. High misalignment angles could induce a dramatic decrease in the count of teeth in contact to less than 40% of the total number [31], resulting in fluctuations in the force exerted on each tooth and even some teeth carrying no load [32,33]. The tooth-side clearance, spline tooth stiffness, transmitted torque, and basic tooth profile parameters all affected the count of actually engaged teeth [34]. Further research discovered that increasing the tooth-side clearance, torque, and dynamic load coefficient could increase the count of engaged teeth, and increasing the stiffness of a single tooth reduces engaged teeth [35]. By reducing the stiffness of the spline teeth and nearby components (such as bearings), more teeth could come into contact at the same time [36]. Furthermore, under the same assembly conditions and manufacturing defects, increasing torque could exacerbate the uneven circumferential load distribution [37].

The aforementioned works concentrated on general spline connections. Furthermore, research on the separator plates‘ spline structure in clutches was mostly focused on thermal stress analysis. Significant temperature gradients could arise in the spline teeth during clutch engagement, resulting in a conspicuous concentration of thermal stress at the tooth roots [38]. The radial thermal stress on dual separator plates was substantially lower than the circumferential thermal stress, which might cause warping and fatigue cracking [39].

In summary, present research on the fracture mechanisms of spline teeth are mostly concerned with the general types of spline pairs (spline shafts and sleeves), with little attention paid to the separator plates’ spline, a typical annular thin plate structure. Although relevant research exists, they are mostly focused on thermal stress, with little exploration of other aspects, particularly tooth count and profile, misalignment error, etc. Furthermore, the recent scholarly assessments of spline contact properties under misalignment situations are primarily qualitative, lacking research on stress distribution properties under various types of misalignment, engagement conditions at specific offset values, and effects of varied tooth profiles and tooth counts on engagement uniformity.

This paper discusses the stress characteristics of the spline teeth of wet multi-plate clutch separator plates, surrounding structural factors (tooth profile and tooth count), and assembly misalignment parameters. The separator plates’ stress distribution properties are investigated using FEM. This work is critical for directing the optimum design of the spline structure in wet multi-plate clutch separator plates.

## 2. Materials and Methods

### 2.1. Analytical Model

Splines are classified into four categories based on tooth profile differences: rectangular, involute, triangular, and trapezoidal. Aside from the triangular splines, which are mostly utilized for light weights and small-diameter static connections, the remaining can all be used in big mechanical transmission systems. As a result, this paper will undertake a comparative analysis and investigation of these three types of splines.

When a multi-plate clutch engages, the control oil enters the piston chamber, overcoming the separation spring’s resistance and pushing the piston axially, therefore eliminating the clearance between friction parts. The torque is conveyed by the friction action between the separator plates and the friction plates. Figure 3 depicts the separator plate’s force situation.

Based on the examination of the forces acting on the separator plate, it is clear that the cylinder liner’s tangential force on the spline teeth is the primary cause of the separator plate tooth fracture (crack). When examining a single spline tooth, the force model is now a conventional cantilever beam model.

In light of the above assumptions, it is easy to obtain the torque’s expression shared by a single spline tooth, $\bar{T}=T_{s}/Z$, where *Z* is the count of teeth in a single separator plate; *T_s_* is the torque transmitted by a single separator plate.

At this point, it is easy to calculate the nominal tangential force:(1)$$F_{t}=\frac{2\;\bar{T}}{d_{w}}$$
where *d_w_* is the average circular diameter of the rectangular spline, the sum of the major and minor diameters (for involute splines, *d_w_* is the pitch circle diameter).

According to the assumptions, *F_t_* is uniformly distributed along the contact surface of the tooth profile and can be regarded as a uniformly distributed load. The shear force and bending moment diagrams can be drawn utilizing the cantilever beam theory, as shown in Figure 3. The maximum tooth root bending moment’s calculation formula is as follows:(2)$$M_{\max}=F_{t}\cdot \frac{h}{2}$$
where *h* is the spline tooth’s total height.

The spline tooth’s stress concentration area is located near the root transition position, for which the formula for calculating the bending section coefficient *W_z_* is as follows:(3)$$W_{z}=\frac{1}{6}\cdot b\cdot S_{F_{n}}^{2}$$
where *b* is the thickness of the separator plate; $S_{F_{n}}$ is the chordal tooth thickness at the stress concentration area. For rectangular tooth splines and trapezoidal tooth splines, $S_{F_{n}}$ is equal to the thickness of the tooth at the root; for involute splines, it must be calculated through the following formula:(4)$$S_{F_{n}}=D_{F_{e}}\cdot \sin\left[\frac{S_{p}}{D}+inv\alpha -inv\left(\mathrm{arc}\cos\frac{D\cdot \cos\alpha }{D_{F_{e}}}\right)\right]$$
where α is the pressure angle; *D* is the pitch circle diameter; $D_{F_{e}}$ is the diameter of the starting circle of the involute curve; $S_{p}$ is the chordal tooth thickness on the reference circle; *inv*(*x*) is the involute function, whose expression is *inv*(*x*) = *tanx-x* (the angle unit is in radians).

Therefore, the expression for bending stress $\sigma_{F,\max}$ is as follows:(5)$$\sigma_{F,\max}=\frac{M_{\max}}{W_{z}}$$

Combining the above calculation results yields the expression for the maximum bending stress at the tooth root as follows:(6)$$\sigma_{F,\max}=\frac{6hT_{s}}{bd_{w}ZS_{F_{n}}^{2}}$$

### 2.2. FE Model

Traditional theoretical calculation approaches have limitations when dealing with non-standard or complex geometric structures, while FEM can solve concerns with complex forms and boundary conditions, providing more accurate findings for stress, strain, and other assessments. This paper’s FE model is built by the ANSYS software (version: 2022R1).

When establishing the FE model, some simplifications need to be made to the clutch geometric structure because of the limited computational capabilities of the workstation and the focus on a single separator plate’s stress distribution: simplifying the cylinder liner’s thread and chamfer structures, piercing it axially while maintaining the integrity of the spline, as shown in Figure 4. With the aforementioned simplifications, studying the interaction of a single separator plate and cylinder liner becomes more natural and straightforward. This paper’s FE models are divided into two categories: static torsion models under uniform load conditions (without misalignment errors) and static torsion models under biased load situations (with misalignment errors).

Assign material attributes to each component. Based on the current application conditions, the separator plate is built of 30CrMnSi, whereas the cylinder liner is formed of ASTM standard steel A36, with the mechanical parameters shown in Table 1.

For the contact areas between the separator plate and the cylinder liner, all are set to frictionless contact, whose contact algorithm is the default program controlled.

A hexahedral 3-D element type is selected because of its higher computational accuracy, utilizing the multi-zone meshing method. The body of influence (BOI) refinement approach is used to refine meshes in the separator plate’s tooth area. Meanwhile, those in the contact area between the two components of the cylinder liner are refined using a separate definition of the contact dimensions.

The mesh density is determined by MIV(mesh independence verification), which can protect the FEM results from mesh size and ensure analysis credibility. MIV entails running a series of analyses using finer mesh density until the findings converge within an acceptable tolerance. As shown in Table 2, when the elements exceed 250,000, the calculation results tend to stabilize. Consequently, the mesh density is decided and the final FE model consists of approximately 310,000 discrete 3-D elements and 380,000 nodes.

## 3. Results and Discussion

### 3.1. Even Load

In this section, the main simulation parameters are shown in Table 3. The typical simulation results about the stress distribution are shown in Figure 5, which demonstrates that the stress-concentrated location on the contact side is near the tooth root.

#### 3.1.1. Tooth Count

Based on the cylinder liner’s inner and outer diameter measurements, under the condition that the ratio of the separator plate key tooth width to the cylinder liner steel post width is constant, four types of dual separator plates with the tooth counts of 12, 18, 24, and 30 are designed, as well as their matching cylinder liners. A simulation analysis is performed on these four assembly combinations, yielding the relationship between the peak stress of the separator plate, the teeth count, and the torque, as shown in Figure 6.

The diagram shows that the separator plate’s max stress value is directly proportional to the transmitted torque. Increasing the teeth count lowers the max stress value, but more is not always better. Under each tooth count condition, the relationship between the maximum stress and torque is nearly linear, implying that the max stress is roughly proportionate to the torque. Furthermore, when transmitting the same torque, the fall in maximum stress from 12 to 30 teeth reduces gradually between the neighboring tooth counts. For a torque of 600 N·m, the stress decreases from 12 to 18 teeth, 18 to 24 teeth, and 24 to 30 teeth are 12.7 MPa, 5.5 MPa, and 2.8 MPa, respectively, with a distinct diminishment. When considering issues like increased production difficulty and assembly faults, having more teeth is not necessarily advantageous. Among the four tooth counts given, 18 or 24 teeth are relatively better options.

#### 3.1.2. Tooth Profile

This section investigates FE simulations on the separator plates with the three tooth profiles (rectangular, trapezoidal, and involute). When designing the separator plates with the three different tooth profiles, the tooth thicknesses (for trapezoidal teeth, the tooth thickness is measured at the pitch circle; for involute teeth, it is measured at the pitch circle) are kept equal while maintaining the same addendum circle and dedendum circle diameters for all three profiles. In the study of the trapezoidal teeth, as shown in Figure 7a, with equal average thickness (tooth thickness at the average circle, whose radius $R_{a}$ satisfies $R_{a}=\frac{R_{t}+R_{r}}{2}$), three sets of independent variables are developed, with the trapezoidal tooth’s pressure angles set at 10°, 20°, and 30°, respectively. The rectangular tooth can be classified as a special trapezoidal tooth with a pressure angle of 0°. Based on the above content, the effect of various pressure angles on the max stress is investigated, of which the results are given in Figure 7b.

The diagram shows that, under the same torque settings, the maximum stress on the separator plates with the rectangular, involute, and trapezoidal teeth falls sequentially. The reduction from rectangular to involute tooth is 17 MPa (27.8%), from rectangular to trapezoidal tooth (with a default pressure angle of 30° unless specified otherwise) is 24 MPa (39.3%), and from involute to trapezoidal tooth is 7 MPa (15.9%). This suggests that both trapezoidal and involute teeth can improve the strength at the tooth root, with the trapezoidal teeth having the better benefit.

Furthermore, for the trapezoidal teeth, as the pressure angle increases, the maximum stress at the tooth root reduces in a roughly linear downward trend. However, a bigger pressure angle does not always mean a better result. According to Figure 8, the contact stress distribution is not uniform, consistent with real-world conditions. As the pressure angle increases, the contact stress at the tooth root falls while the contact stress at the tooth top rises. At a pressure angle of 30°, the contact stress at the tooth tip exceeds that at the tooth root for the first time. Excessive contact stress at the tooth tip, as opposed to the root, is more likely to result in failure forms such as pitting and flaking on the tooth surface. As a result, the pressure angle should not exceed 30°. Among the pressure angle groups indicated, 20° is a better choice.

### 3.2. Biased Load (Misalignment)

To mimic the misaligned installation between the separator plate and the cylinder liner, a certain angular or radial displacement is specified during component assembly, followed by the FE simulation studies using this configuration. Referring to the misalignment deviation range within the assembly tolerance where no interference occurs and Kou’s quantitative calculation method for misalignment offset [4], this paper sets the angular offset to 0.04°, 0.08°, and 0.12°, and the radial offset to 0.02 mm, 0.04 mm, 0.06 mm, and 0.08 mm. The main simulation parameters for this section are shown in Table 4.

#### 3.2.1. Angular Misalignment

The misalignment parameters for the angular misalignment condition are 0 (blank control), 0.04°, 0.08°, and 0.12°, while the structural parameters include the tooth count (12 teeth, 18 teeth, and 24 teeth) and tooth profile (rectangular, involute, and trapezoidal).

Figure 9 displays the simulation findings for the stress distribution and deformation characteristics under the angular misalignment situations (using an 18-tooth rectangular tooth profile with a 0.08° offset as an example).

It is worth mentioning that the stress distribution among the teeth on both sides of the separator plate is trigonometric with the peaks and valleys aligned, where the phase angles of the distribution functions differ by 180°.

Under angular misalignment, the stress distribution has the following properties: The max stress occurs at the top site on the contact side of the tooth. The stress distribution among all the teeth of the separator plate is trigonometric, as shown in Figure 10 and Figure 11. The deformation features are as follows: Warping deformation takes place at both ends of the separator plate, with the warping directions at the two ends being opposed (demonstrating centrally symmetric qualities). Tooth edge curling takes place on the side that experiences a larger force. This is clearly shown in Figure 9, and it corresponds to the stress distribution features.

Furthermore, simulation tests were carried out on the separator plates with varying teeth counts and profiles under different angular misalignment circumstances. Figure 10 and Figure 11 show the results of the stress distribution among all the teeth.

The unevenness coefficient of the stress distribution for each separator plate is defined as the standard deviation of the stress across the teeth. A larger unevenness coefficient indicates more uneven stress distribution among the spline teeth, while a lower coefficient indicates a more uniform stress distribution. Figure 12 shows how the stress distribution unevenness coefficient and peak stress vary with the tooth count and angular offset.

As the teeth count rises, the peak stress on the separator plate lowers, diminishing the unevenness of the stress distribution over the teeth. As shown in Figure 12a, when the angular misalignment is less than 0.04°, the unevenness coefficients for the three tooth counts are nearly identical. When the misalignment exceeds 0.04°, changes in the unevenness coefficient progressively appear across the three tooth counts (albeit the difference is not so significant), indicating that having more teeth results in less unevenness. In terms of peak stress, as shown in Figure 12b, for the same teeth count, the peak stress increases with the offset; for the same offset, there is an overall trend where more teeth lead to lower peak stress on the separator plate.

Under the same load conditions, from rectangular to trapezoidal to involute tooth, the peak stress on the separator plate progressively reduces, meanwhile, the unevenness gradually decreases. As shown in Figure 13a, for the same offset, the unevenness coefficient shows a decreasing trend from rectangular to trapezoidal to involute on the whole, although the trapezoidal tooth increases little compared to the involute tooth at 0.04° offset. As illustrated in Figure 13b, the peak stress steadily reduces from rectangular to trapezoidal to involute, although the increase is not large. In conclusion, under the angular misalignment conditions, both trapezoidal and involute teeth can improve the strength of the separator plate, with involute providing better improvement.

#### 3.2.2. Radial Misalignment

The misalignment parameters chosen for the radial misalignment section are 0 (blank control), 0.02 mm, 0.04 mm, 0.06 mm, and 0.08 mm, whereas the structural parameters are tooth count (12 teeth, 18 teeth, and 24 teeth) and tooth profile (rectangular tooth, involute tooth, and trapezoidal tooth). Figure 14 depicts the simulation findings for the separator plate’s stress distribution and deformation characteristics under the radial misalignment situations (using an 18-tooth rectangular tooth profile with an offset of 0.08 mm as an example).

Under radial misalignment situations, the stress distribution has the following characteristics: Only a small part of teeth transmit torque, while the rest remain unloaded. The highest stress occurs at the tooth root, and the stress distribution on both sides of the separator plate is nearly symmetrical. The deformation features are as follows: The separator plate’s annulus transforms into an elliptical ring, warping (asymmetrical) occurs near the loaded teeth, and the loaded teeth undergo bending deformation. These characteristics are notable in Figure 14, which differ significantly from that of the angular misalignment.

Continuing, simulation tests were carried out on the separator plates with varying tooth counts under varied radial misalignment situations. With a similar approach, the stress distribution diagrams of each tooth can be drawn, as shown in Figure 15; in the meantime, the stress distribution unevenness coefficient and peak stress with changes in the tooth count and radial misalignment are shown in Figure 16.

When radial misalignment occurs, the unevenness in the stress distribution among the teeth quickly rises due to the rapid reduction in the count of teeth carrying the load. In contrast to angular misalignment, where each spline tooth bears force (albeit unevenly), radial misalignment causes most teeth not to participate in the torque transmission, as shown in Figure 16b, resulting in 3~4 times the maximum stress on the loaded teeth. For the same teeth count, the larger the radial offset, the fewer teeth are meshing. With the same offset, the more teeth the separator plate has, the more teeth are loaded.

As illustrated in Figure 16a,b, a similar trend is seen for the unevenness coefficient and peak stress: for each group of radial offset, the unevenness coefficient and peak stress for 18 and 24 teeth are quite comparable to one another and are both lower than that of 12 teeth. The unevenness coefficient and peak stress values of 18 and 24 teeth are similar because for the separator plates with a larger number of teeth, the width of the key teeth is smaller, resulting in lower strength under the same load. However, as shown in Figure 17a, under the radial offset conditions, although the count of loaded teeth increases, the actual counts of the loaded teeth for the three sets do not differ much. This reduces the load borne by each meshing tooth (increased strength), which offsets the decrease in strength caused by the reduction in tooth width due to the increase in the design teeth count.

Consequently, it can be concluded that under radial offset load, correctly increasing the teeth count can considerably improve the strength of the separator plate while also improving the uniformity of the load distribution among the teeth. However, once the teeth count surpasses a specific threshold, continuing to increase it provides no extra benefit.

Additionally, simulation investigations were carried out with varied tooth profiles under various radial misalignment situations, sampling the same area on each tooth, resulting in Figure 18. The stress distribution unevenness coefficients with changes in the tooth profile and radial misalignment are illustrated in Figure 19.

Figure 19a,b show that the change trend for the trapezoidal teeth and involute teeth is basically the same: in each group of radial offset, the unevenness coefficient and peak stress values of the trapezoidal and involute teeth are very close (slightly higher for the involute teeth), both of which are less than that of the rectangular teeth. They show a considerable increase from 0 to 0.02 mm, but the increase decreases dramatically after 0.2 mm, with a gradually more pronounced reduction as compared to the rectangular teeth.

In Figure 17b, the actual count of the loaded teeth is quite similar for the trapezoidal and involute teeth at identical offset settings, and both are larger than the rectangular teeth.

## 4. Conclusions

This study investigates the spline tooth fracture (cracking) of separator plates in wet multi-plate clutches, considering the tooth count, tooth profile, and installation defects (misalignment errors). A FE model is developed to mimic the interaction between the separator plates and the cylinder liners, yielding the following results:Increasing the teeth count can effectively reduce the separator plate’s max stress. Specifically, for the trapezoidal teeth, raising the pressure angle suitably can significantly improve their strength.Compared to the rectangular teeth, both the trapezoidal and involute teeth can improve the strength at the tooth root, with the trapezoidal teeth having the better effects.On angular misalignment conditions, all the teeth are loaded, the stress on each tooth exhibiting a trigonometric distribution, whereas in radial misalignment cases, only some teeth are loaded, of which the peak stress and unevenness are much greater than that of angular misalignment.Under misalignment situations, increasing the tooth count or modifying the tooth design from rectangular to trapezoidal or involute can minimize the max stress, causing a more even stress distribution among all the teeth.

## Figures and Tables

**Figure 1 materials-17-03039-f001:**
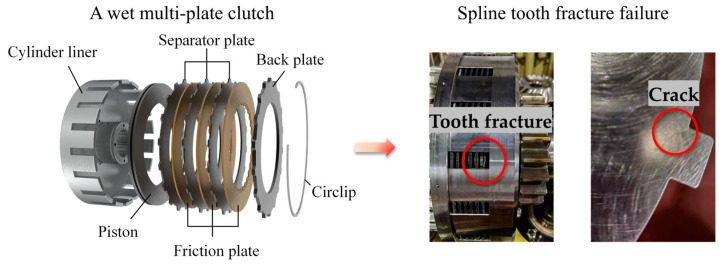
A wet multi-plate clutch and the separator plates’ tooth fracture and crack.

**Figure 2 materials-17-03039-f002:**
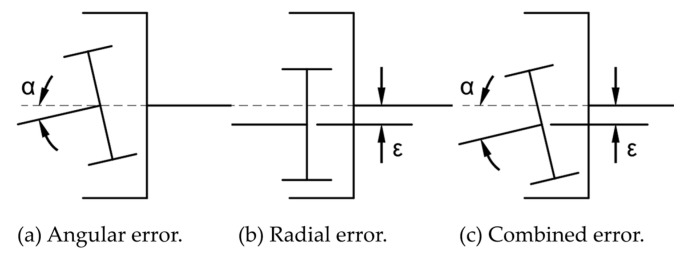
Misalignment error model.

**Figure 3 materials-17-03039-f003:**
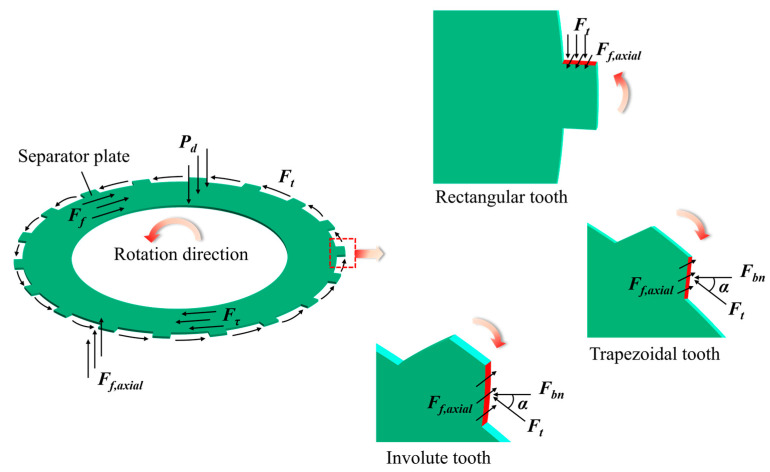
Force diagram for the separator plate and single teeth.

**Figure 4 materials-17-03039-f004:**
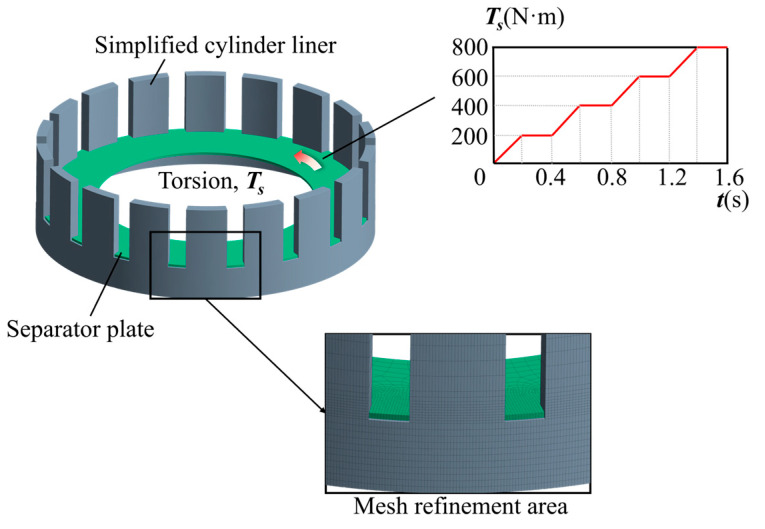
FE model.

**Figure 5 materials-17-03039-f005:**
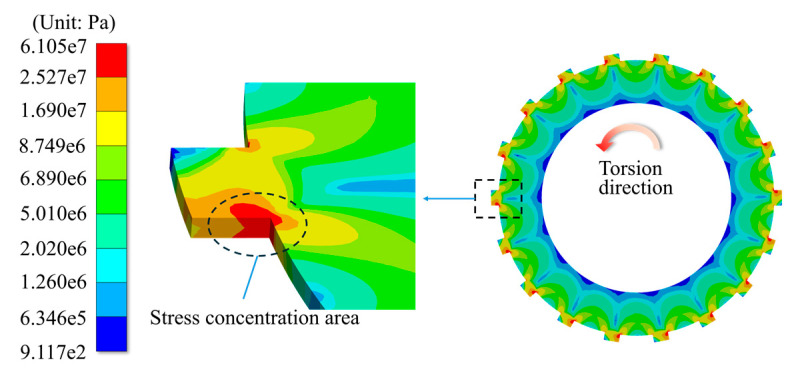
Stress distribution results under even load conditions.

**Figure 6 materials-17-03039-f006:**
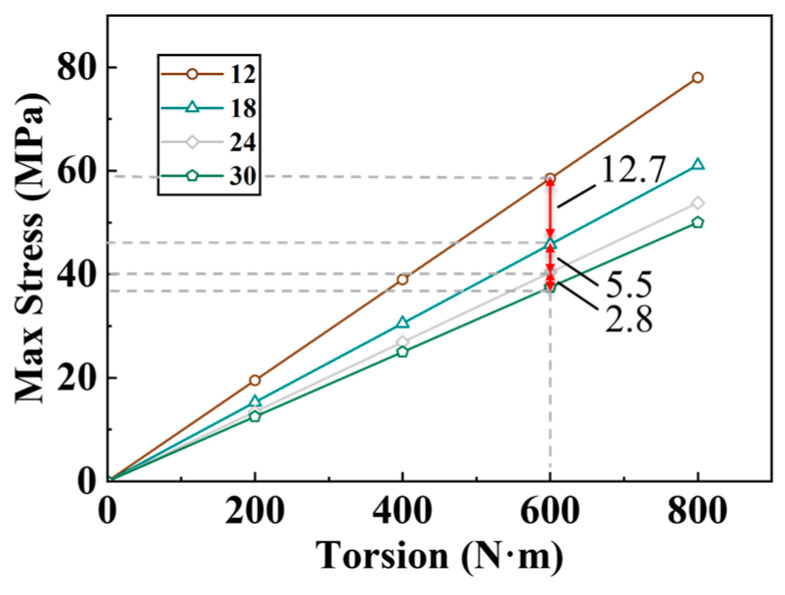
Max stress results for torque and tooth count variables.

**Figure 7 materials-17-03039-f007:**
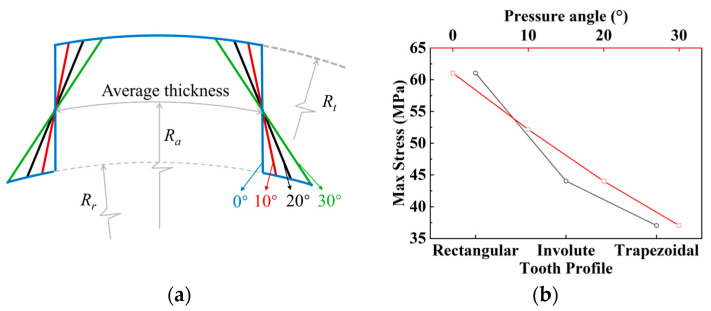
Simulation results of trapezoidal teeth with varying pressure angles: (**a**) trapezoidal profiles with different pressure angles; (**b**) max stress results for tooth profile variables.

**Figure 8 materials-17-03039-f008:**
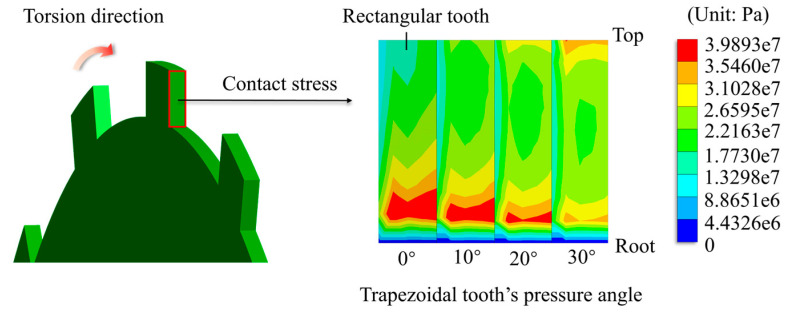
Comparison diagram of contact stress for different trapezoidal teeth.

**Figure 9 materials-17-03039-f009:**
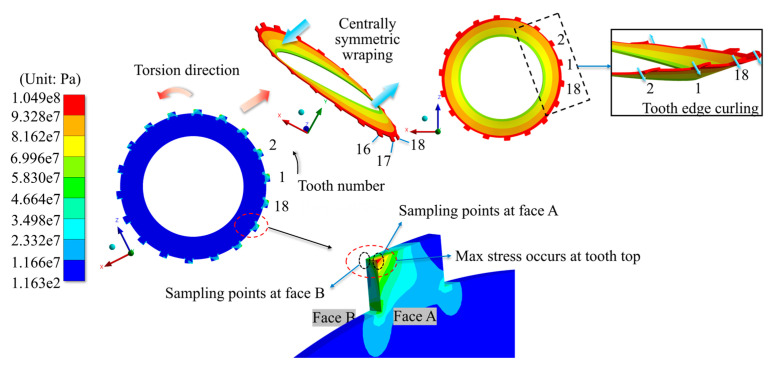
Equivalent stress distribution and deformation characteristics under angular misalignment.

**Figure 10 materials-17-03039-f010:**
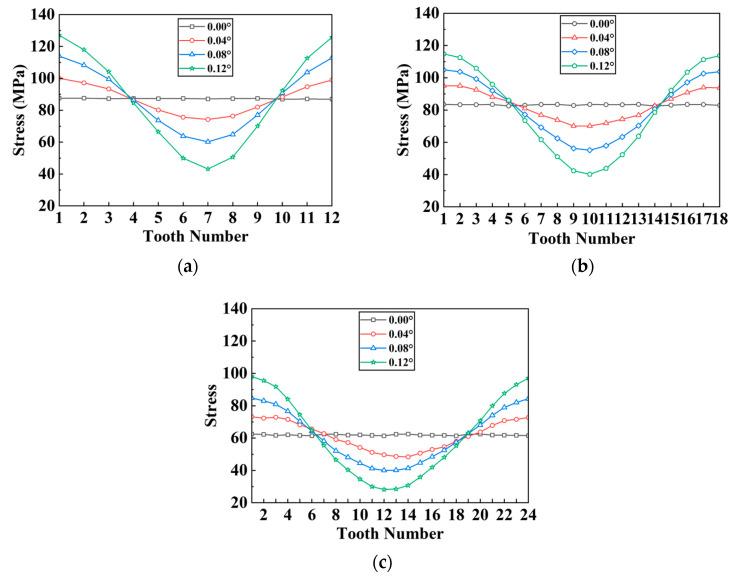
Stress distribution of each tooth on separator plates under different angular misalignment conditions and tooth counts: (**a**) 12 teeth; (**b**) 18 teeth; (**c**) 24 teeth.

**Figure 11 materials-17-03039-f011:**
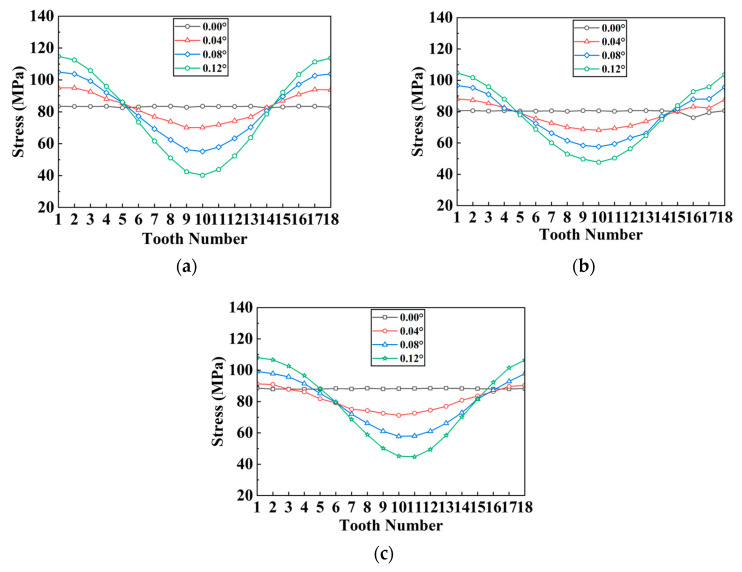
Stress distribution of each tooth on separator plates under different angular misalignment conditions and tooth profiles: (**a**) rectangular tooth; (**b**) involute tooth; (**c**) trapezoidal tooth.

**Figure 12 materials-17-03039-f012:**
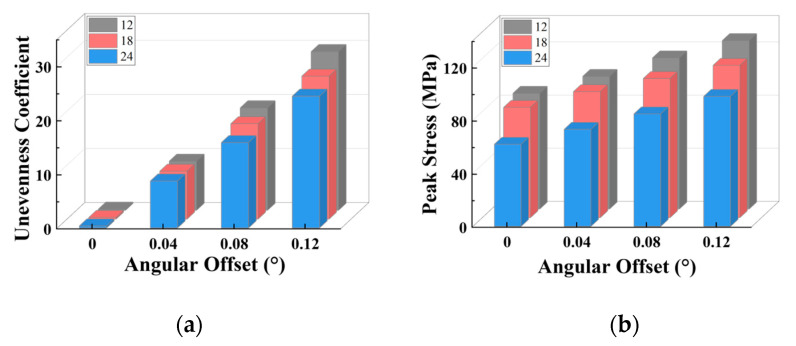
Variation characteristics of the unevenness coefficient and peak stress under different tooth counts: (**a**) unevenness coefficient; (**b**) peak stress.

**Figure 13 materials-17-03039-f013:**
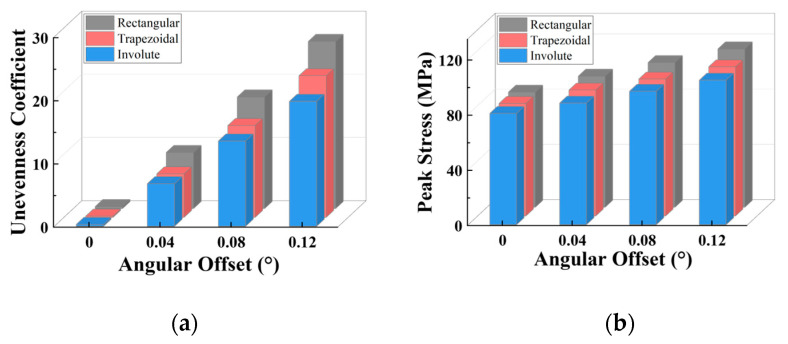
Variation characteristics of the unevenness coefficient and maximum stress under different tooth profiles and angular offset: (**a**) unevenness coefficient; (**b**) peak stress.

**Figure 14 materials-17-03039-f014:**
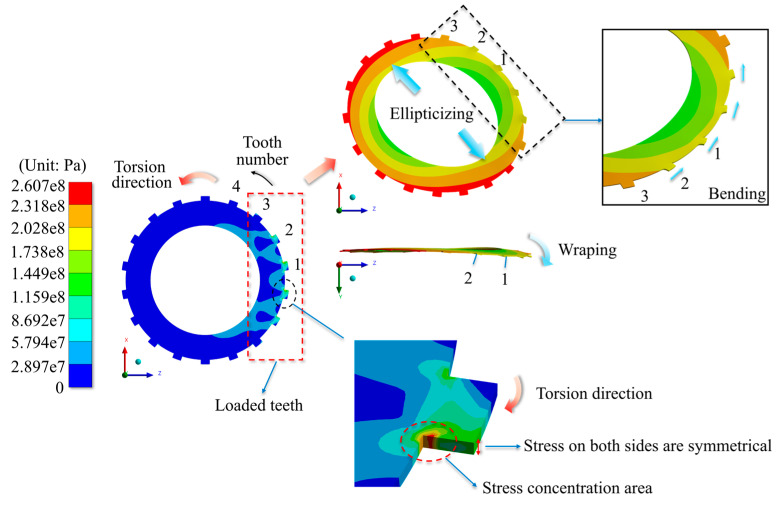
Equivalent stress distribution and deformation characteristics under radial misalignment.

**Figure 15 materials-17-03039-f015:**
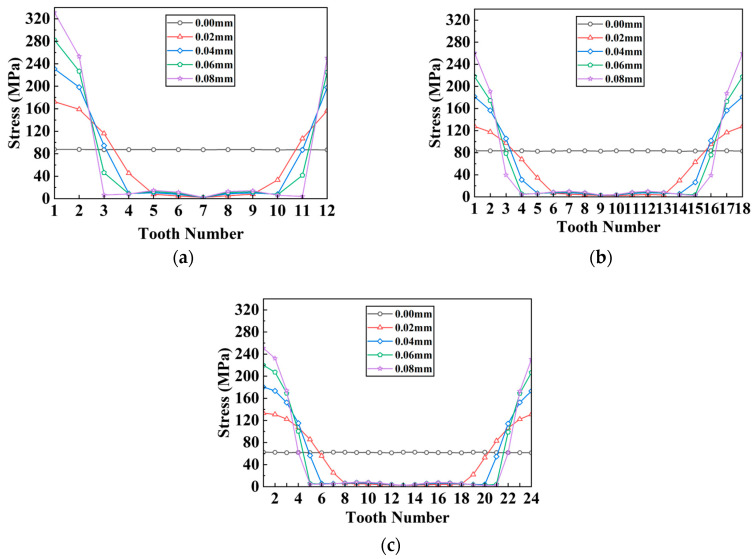
Stress distribution of each tooth on separator plates under different radial misalignment conditions and tooth counts: (**a**) 12 teeth; (**b**) 18 teeth; (**c**) 24 teeth.

**Figure 16 materials-17-03039-f016:**
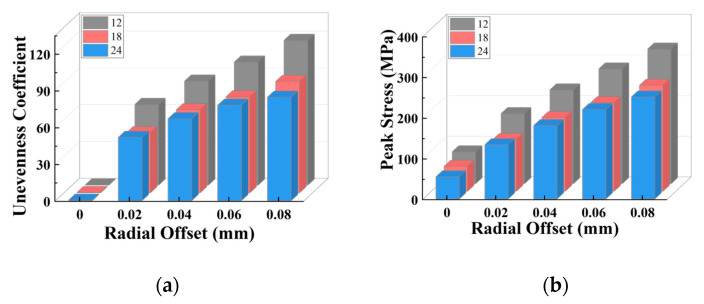
Variation characteristics of the unevenness coefficient and maximum stress under different tooth counts: (**a**) unevenness coefficient; (**b**) peak stress.

**Figure 17 materials-17-03039-f017:**
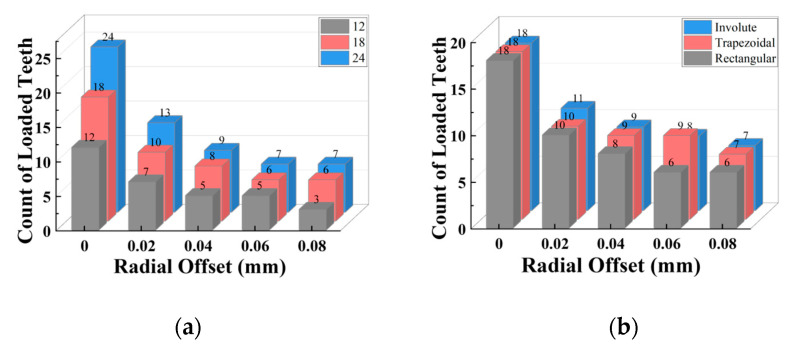
Variation characteristics of the count of the loaded teeth under radial misalignment and different tooth counts and profiles: (**a**) teeth count; (**b**) teeth profile.

**Figure 18 materials-17-03039-f018:**
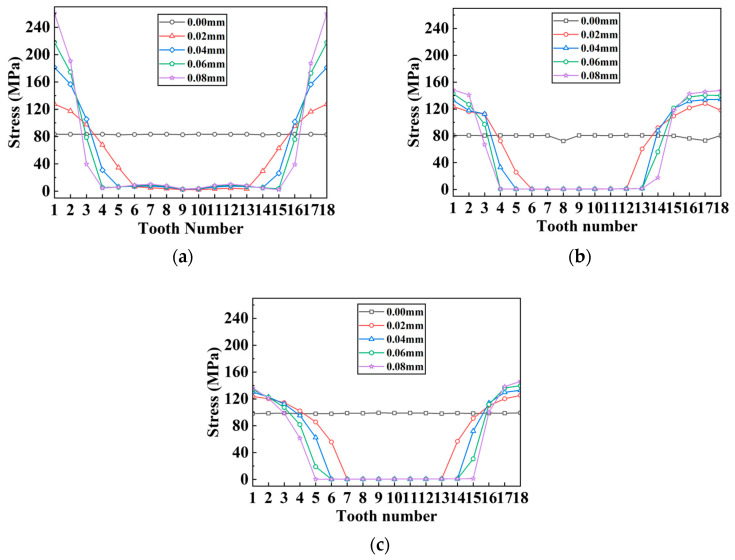
Stress distribution of each tooth on the separator plate under different radial misalignment conditions and tooth profiles: (**a**) rectangular tooth; (**b**) involute tooth; (**c**) trapezoidal tooth.

**Figure 19 materials-17-03039-f019:**
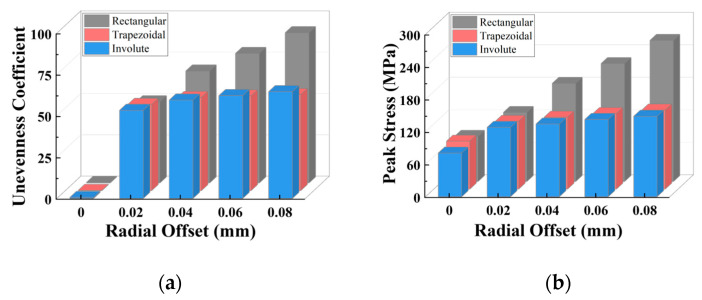
Variation characteristics of the unevenness coefficient and maximum stress under different tooth profiles and radial offset: (**a**) unevenness coefficient; (**b**) peak stress.

**Table 1 materials-17-03039-t001:** Mechanical properties of the two steels.

Steel Type	Elastic Modulus (GPa)	Poisson’s Ratio	Density (Kg/m^3^)	Strength (MPa)	
				Compressive/Tensile Ultimate	Compressive/Tensile Yield
30CrMnSi *	216	0.3	7750	1080	835
A36	200	0.3	7850	460	250

* the data for 30CrMnSi are sourced from Cao [40] and GB/T 3077-2015 [41].

**Table 2 materials-17-03039-t002:** The process of MIV.

Elements	Nodes	Calculation Time (min)	Max Equivalent Stress	
			Value (KPa)	Relative Deviation from the Previous Mesh (%)
176,797	224,525	133	17,533	-
213,462	278,901	179	19,957	13.825
259,297	320,178	205	21,097	5.712
310,269	384,113	253	21,275	0.844
413,120	502,072	328	21,358	0.390

**Table 3 materials-17-03039-t003:** Main simulation parameters under even load condition.

Factors	Parameters			
Tooth count	12	18	24	30
Tooth profile	rectangular	trapezoidal	involute	-
Pressure angle *	0°	10°	20°	30°

* the pressure angle refers to the trapezoidal teeth.

**Table 4 materials-17-03039-t004:** Main simulation parameters under biased load condition.

Factors	Parameters				
Tooth count	12	18	24	-	-
Tooth profile	rectangular	trapezoidal	involute	-	-
Angular offset	0°	0.04°	0.08°	0.12°	-
Radial offset	0 mm	0.02 mm	0.04 mm	0.06 mm	0.08 mm

## Data Availability

The original contributions presented in the study are included in the article, further inquiries can be directed to the corresponding author.

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
