# Peer review of "Study on Spline Stress of Separator Plates in a Wet Multi-Plate Clutch"

_materials, 2024, doi:10.3390/ma17123039_

Round 1

Reviewer 1 Report

Comments and Suggestions for Authors

The authors presented an article “Study on spline Stress of Separator plates in a Wet Multi-plate Clutch”, I appreciate the effort you have put into conducting this research and compiling the findings. However, the following points needs to be addressed to improve the quality and impact of the manuscript:

1.      What is the need of this research? Will the production of trapezoidal or involute teeth is commercially possible? If cannot what is the need of them?

2.      The simulation parameters need to be defined in a tabular form?

3.      “spline tooth's dangerous section”, how can you define this?

4.      Why same reference has been cited twice in Line 176?

5.      Why “hexahedral element” type only? What are the shortcomings of other element type?

6.      0.04°, 0.08°, and 0.12°, how these angular misalignments are selected?

7.      Error! Reference source not found.. whats this? Check in line 314, 324, 335,

8.      There are many writing errors in the manuscript? Remove them?

Comments on the Quality of English Language

moderate english editing is required

Reviewer 2 Report

Comments and Suggestions for Authors

Comments and suggestions are provided in the attached file.

Reviewer 3 Report

Comments and Suggestions for Authors

Line 33: Circumferential and axial, wrong font or font size used.

Line 115: The type of clutch as shown in Figure 1 is actuated by an external lever pushing a thrust bearing against the clutch pack. "Control oil", or control air, is for clutches usually fitted to tool machines, with the control medium supplied via a hollow shaft.

Line 146: Check your indexes. the "p" in Sp should not be italic as it is an index. Same goes for Fmax in line 151 and beyond.

Line 156: Judging from Chapter 2.2, you have limited your testing to just two plate spline teeth, however later in line 183 it is seen that 3-D elements are used, while from Chapter 2.2. one might conclude from Chapter 2 that 2-D elements are used.

Also, it is inconclusive whether proper contact mechanics are used to apply the load to the spline flanks.

Line 207: You mention rectangular, involute and trapezoidal spline teeth. Can you mention some standards according to which the geometry was generated, as otherwise your research cannot be reproduced?

Line 248: The graph at the bottom right of Figure 9 should be broken out into a separate figure or removed completely.

Line 315: Reference source missing, same in Line 324 and 335.

Line 384: Please resolve the funding as appropriate for your article.

The English language is fine. Nothing beyond a regular spelling and grammar check is needed.

Round 2

Reviewer 1 Report

Comments and Suggestions for Authors

The reviewer is satisfied with the explanation provided by the authors to the queries raised.

Comments on the Quality of English Language

minor checking is required